# Direct and Stress-Buffering Effects of COVID-19-Related Changes in Exercise Activity on the Well-Being of German Sport Students

**DOI:** 10.3390/ijerph18137117

**Published:** 2021-07-02

**Authors:** Laura Giessing, Julia Kannen, Jana Strahler, Marie Ottilie Frenkel

**Affiliations:** 1Institute of Sports and Sports Sciences, Heidelberg University, 69120 Heidelberg, Germany; julia.kannen@gmx.de (J.K.); marie.frenkel@issw.uni-heidelberg.de (M.O.F.); 2Faculty of Psychology and Sport Science, Justus-Liebig University Gießen, 35394 Gießen, Germany; Jana.Strahler@psychol.uni-giessen.de

**Keywords:** pandemic, stress, coping, mental health, mood, emotion regulation, physical activity

## Abstract

Maintaining or initiating exercise activity in the COVID-19 pandemic may act as a buffer against the observed stress-related deterioration in well-being, with emotion regulation (ER) discussed as a possible moderator. Therefore, the present study investigated the interaction between stress, exercise activity (EA), and ER on mood. In an online survey, 366 German sports science students (56% women, *M*_age_ = 23.04, *SD* = 2.87) reported their stress levels (general and COVID-19-specific), mood (energy, valence, calmness), EA before and during the pandemic, and use of ER strategies in spring 2020. Pandemic-related change in EA was calculated as residual change. Due to gender differences in mental health and EA, the main and interaction effects were tested in twelve hierarchical regression analyses, separately for men and women. Overall, EA significantly decreased during the pandemic and was positively associated with energy in both men and women. ER was positively associated with women’s energy, but negatively with all three mood dimensions in men. Only one three-way interaction appeared significant: in the case of high stress, low levels of EA and high use of ER were associated with the greatest deteriorations in energy in men. Our findings suggest that EA may buffer deteriorations in energy in men with high stress and difficulties in ER.

## 1. Introduction

In spring 2020, the emergence of the COVID-19 pandemic resulted in unprecedented public health measures disrupting daily life routines, which might have caused the pandemic to be a large-scale stressor for many individuals. Although the “stay-at-home” mandates and physical isolation may have had a positive effect in mitigating the virus transmission, a growing body of literature suggests that immediate and potentially long-lasting negative psychological effects, such as high levels of stress, anxiety, and depression, may have resulted from these restrictive measures (for reviews see [1,2,3]). Given the well-documented direct and stress-buffering effects of physical exercise on mental health [4,5,6], maintaining or initiating exercise activity despite the governmental restrictions may have been an important coping mechanism to counteract stress-related deteriorations in well-being during the COVID-19 pandemic [7].

Importantly, the COVID-19 pandemic and related public measures have not only impacted mental health, but also exercise engagement itself. The reinforced lockdown restrictions (e.g., closing of gyms and fitness clubs, restricted access to parks and outdoor environments) have limited possibilities to engage in structured or outdoor exercise, requiring people to be innovative in their exercise activities. Most studies have reported an overall decrease in exercise engagement during the pandemic (e.g., [8]). Some researchers have warned that the forced stop period might result in detraining, related physiological training losses [9], and negative psychological effects [10,11,12,13,14], particularly among highly active individuals who are known to experience severe deteriorations of well-being in response to exercise deprivation [15,16]. However, more detailed observations show that often, a significant proportion of the samples was able to maintain or even increase their exercise engagement during the current pandemic [12,13,17]. Although athletes spent less overall training time each week and conducted shorter training sessions, they performed more strength training, engaged in more high-intensity training [18], and could maintain their fitness levels [19]. For those, adapted exercise activity might be counted as a defense strategy and a preventive measure. For instance, home-based exercise can help to regulate the immune system and delay immunological aging in both non-clinical populations and COVID-19 patients [20,21]. Likewise, regular physical activity was associated with less distress and better well-being during the COVID-19 pandemic [8,12,13].

Physical exercise, defined as planned and structured bodily movements that are conducted to increase physical fitness [22], has been robustly found to contribute to well-being by improving positive and reducing negative affective states (for reviews see [4,23]). Affective states—as a core component of well-being [24]—are valenced feelings (pleasure vs. displeasure), which express themselves in emotions and moods. Emotional states are generally short-lived, intense responses to identifiable stimuli (i.e., anxiety, anger, surprise), whereas moods (i.e., irritation, activation, calmness) are often less intense, last longer and do not necessarily have an identifiable stimulus [25]. Physically active individuals generally experience less stress, depression, and anxiety [26,27]. Simultaneously, exercise contributes to the experience of positive affect, well-being, and happiness [27,28,29,30]. For instance, healthy subjects reported feeling better and more energized promptly after self-reported [31] and objectively assessed physical activity [32]. Moreover, the association between exercise and well-being is particularly strong in individuals who are faced with stressful circumstances. That is, exercise may act as a coping resource or “buffer” against the detrimental effects of stress on well-being, i.e., exercise attenuates the negative impact of stress on well-being [5,6]. In experience sampling studies, higher exercise levels buffered the negative effects of stressful life events or other stress experiences on well-being [33,34,35,36]. Likewise, intervention studies confirm exercise as a buffer of stress responses [37,38,39,40]. Given the proportions of samples who maintained or increased their exercise engagement during the COVID-19 pandemic [12,13,19], it can be speculated that individuals might adaptively adjust their exercise engagement according to the current stressors or their perceived stress levels. Following this argumentation, the change in exercise engagement should act as a stronger predictor in the relationship between stress and well-being than current exercise activities. Therefore, the lockdowns in response to the COVID-19 pandemic provided a unique opportunity to investigate the effects of changes in exercise activity on well-being when facing a major stressor.

Different mechanisms have been proposed to underlie the stress-buffering effects of regular exercise activity. On the physiological level, habitual exercise is assumed to lead to adaptations in the cardiovascular and cortisol systems, which may improve the efficiency of the stress response, resulting in lower reactivity or faster recovery from stressful events (e.g., [41,42,43]). On the psychological level, one hypothesis holds that exercise buffers against difficulties regarding emotion regulation when facing a stressor [44,45,46,47], potentially mediated by improvements in executive control during exercise [48]. Emotion regulation, as an explicit or implicit process to influence the occurrence, experience, and expression of emotions [49], plays a key role in determining whether stressful life events result in mental health problems [50]. Although various emotion regulation strategies have been conceptually and empirically distinguished [51], the idea of “coactive” emotion regulation suggests that people may employ multiple strategies simultaneously when confronted with stressful events (on average seven strategies [52,53,54]). At the same time, the distinction between adaptive and maladaptive emotion regulation has been abandoned, as adaptiveness seems to be the result of the variability and flexibility in choosing strategies based on situational demands [55,56,57]. Exercise may lessen the impact of stress responses by improving individuals’ ability to emotionally recover from stress, regardless of the employed strategy [44,45,46].

Notably, there are large gender differences in both exercise activity and mental well-being, suggesting that the interaction effects of stress, exercise activity, and emotion regulation on well-being might differ between men and women [58]. Across decades and cultures, men are consistently reported as being more physically active than females, engaging in a greater amount and higher intensity of exercise activity (e.g., [59,60]). In a parallel vein, a robust body of literature shows that women report worse mental health than men (e.g., higher prevalence for depression [61]). Although the evidence of women’s lower positive mental health is less consistent, studies have found better positive mental health in men than women (e.g., [62]). Similar patterns have also been observed during the COVID-19 pandemic, with women engaging in less exercise (at least in high-intensity exercise [13,63]) and demonstrating higher stress, anxiety, and depression and lower well-being [3,63,64,65,66]. These gender differences in well-being might be, in part, explained by gender differences in emotion regulation [67,68], with women using more strategies and implementing these strategies more flexibly than men [69].

Thus, the overall aim of the present study was to investigate exercise as a protective coping resource and its underlying mechanisms during the COVID-19 pandemic, while taking into account gender differences. By placing a specific focus on the change in exercise activity, the present study examined whether exercise contributes to well-being through direct effects, stress-buffering mechanisms or by facilitating emotion regulation. To this end, we tested the three-way interaction between perceived stress, exercise activity, and emotion regulation on mood, separately for men and women. First, perceived stress was assumed to have a negative effect on mood. Second, we expected an overall decrease in exercise activity during the pandemic [8], which has been shown to have a negative impact on well-being [11,12,13,14,70,71,72,73]. Given the beneficial effects of exercise activity on well-being, we expected that current exercise activity and an increase in exercise activity would be positively associated with mood [23] and with higher use of emotion regulation strategies [44,45,46,47]. Third, we expected that exercise activity (current and change) and emotion regulation would moderate the relationship between perceived stress and mood (cf. [5,6,50,74]), in the sense that in the case of low stress, the relationship between exercise or emotion regulation would be relatively weak, whereas in the case of high stress, participants with higher engagement in exercise activity and more emotion regulation would report less mood deterioration compared to participants with low engagement in exercise activity and emotion regulation. Fourth, in line with the previously mentioned research on stress, exercise activity, and emotion regulation, we expected that participants with low exercise activity and low emotion regulation would experience the highest mood deterioration when they are faced with high stress.

## 2. Materials and Methods

### 2.1. Participants

Overall, 366 sports science students from six universities in Baden-Württemberg, Germany, participated in the online survey (SoSci Survey) between 17 April 2020 and 24 April 2020. Participants were recruited through mailing lists of sport institutes at universities in Baden-Württemberg and via social media (e.g., Facebook groups, Instagram, and WhatsApp). They were between 18 and 39 years old (*M*_age_ = 22.89, *SD* = 2.87). The sample consisted of 160 men and 205 women. One participant identified as non-binary and had to be excluded for the gender-segregated analyses.

All study procedures followed the principles of the Declaration of Helsinki. Informed written consent was obtained from all participants. They received no financial compensation.

### 2.2. Measures

Perceived general stress was measured using the German version of the Perceived Stress Scale (PSS-10 [75]; German version [76]). Participants reported the degree to which situations in their life had been unpredictable, uncontrollable, and overloaded in the past month on a five-point Likert scale (0 = never, 1 = almost never, 2 = sometimes, 3 = fairly often, 4 = very often). After reversing the scores on the four positively stated items (Items 4, 5, 7, and 8), a PSS-10 total score was obtained by summing up all 10 items. Higher scores indicated a higher level of perceived general stress. Klein et al. [76] reported good internal consistency (α = 0.84) and construct validity of the PSS-10. In the present study, the internal consistency was good with α = 0.84.

COVID-19-related stress was measured using four self-drafted items. Participants rated the extent of how stressful, challenging, controllable, and threatening they perceived the COVID-19 crisis to be on a scale from 1 (not at all) to 7 (very). A total average score was calculated with one item being reverse coded (i.e., controllable). The internal consistency was α = 0.69.

Mood was measured using the six-item short version of the German Multidimensional Mood Questionnaire [77]. The items represent three bipolar scales, namely valence (V), energy (E) and calmness (C) (content–discontent (V−), tired–awake (E+), full of energy–without energy (E−), unwell–well (V+), agitated–calm (C+), relaxed–tense (C−)). Each item was rated on a seven-point scale ranging from 1 to 7. Wilhelm and Schoebi [77] reported good structural validity, sensitivity to change and reliability for this short scale. For analyses, data from three items (i.e., V−, E−, C−) were reverse coded. Average scores were calculated for valence, energy and calmness.

Exercise activity before and during the pandemic was measured using the Measurement of Daily Activities and Exercise Questionnaire (Bewegungs- und Sportaktivität Fragebogen; BSA-F [78]). Participants named a maximum of three exercise activities they had regularly engaged in within the last four weeks (see Table 1) and indicated the frequency and duration per episode in minutes for each activity. Subjects completed these questions retrospectively for both a four-week period before baseline restrictions and a four-week period during baseline restrictions. For each exercise activity, the total duration of exercise activity (in min/week) was determined by multiplying frequency by duration and—as participants reported the monthly frequency—dividing it by four. We then added up all single durations to obtain a total exercise activity index value.

The extent of emotion regulation use was assessed using six items, each representing one emotion regulation strategy [79]: “I have calmly reflected on my feelings” (reflection), “I have changed the way I think about what causes my feelings” (reappraisal), “I couldn’t stop thinking about my feelings” (rumination), “I have talked about my feelings with others” (social sharing), “I have avoided expressing my emotions” (expressive suppression), and “I have engaged in activities to distract myself from my feelings” (distraction). Each item was rated on a seven-point scale ranging from 1 (not at all) to 7 (very). A total score was calculated by summing up all items, ranging from 6 to 42.

### 2.3. Statistical Analyses

To examine gender differences, a series of independent *t*-tests were computed on all variables. All following analyses were run separate for male and female participants. To examine the change in sports activity, duration of sports activity before and during the pandemic was compared using a *t*-test for dependent samples. Pearson product moment correlation coefficients were used to examine bivariate associations between the predictor, moderator, and outcome variables. Gender differences in the bivariate associations between variables were tested using Fisher’s *z.*

Three-stage regression analyses were performed to determine whether the interaction of perceived stress, exercise activity, and emotion regulation predicted mood (see Figure 1). In sum, four separate regression equations were computed to test the influence of the two distinct stress indicators (general stress, COVID-19-related stress) and the two exercise activity variables (during the lockdown and COVID-19-related change). Emotion regulation was included in all models. The three mood dimensions (energy, valence, calmness) were tested separately, resulting in 12 models per gender. Stress, exercise activity, and emotion regulation were entered as predictors in the first step in each regression, the two-way interaction terms of stress, exercise activity and emotion regulation in the second, and the three-way interaction term of stress, exercise activity, and emotion regulation in the third step. All predictors were mean centered before interaction terms were calculated. To interpret significant two- and three-way interactions, the results of the regression analysis were plotted with high scores corresponding to values + 1 SD and low scores to values − 1 SD (http://www.jeremydawson.co.uk/slopes.htm, accessed on 12 March 2021). Moreover, for significant interactions, simple slope analyses were performed to empirically test which of the pairs of slopes significantly differed from each other [80].

All statistical analyses were performed using SPSS26 (IBM, Chicago, IL, USA). Statistical significance followed conventional criteria with *p*-values < 0.05 considered as being significant.

## 3. Results

### 3.1. Descriptive Statistics and Gender Differences

One participant was excluded from the analyses due to implausible values in the BSA. Table 1 displays the descriptive statistics for all predictor, moderator, and outcome variables. Participants showed higher perceived general stress levels compared to the age- and gender-matched norm values (male: *M* = 12.01, *SD* = 6.51; female: *M* = 13.34, *SD* = 6.75 [76]). Before the pandemic, 179 participants (49.2%) did not attain the recommendations of [81] for moderate-intensity aerobic physical activity (min. 150 min/week). During the lockdown, 215 participants (59.1%) did not attain the recommendations. Compared to norm values, the sample can be considered as moderately active (i.e., 120 to 360 min/week; cf. [43]).

No significant gender differences in exercise activity could be found (see Table 2). Women experienced more general as well as COVID-related stress than men (both *p* < 0.001). Regarding mood, women reported more energy (*p* = 0.03) and felt less calm than men (*p* = 0.01), but did not differ in valence. Women engaged in more emotion regulation than men (*p* < 0.001).

Women engaged in significantly less exercise activity during the pandemic than before the pandemic, *t*(204) = 2.82, *p* = 0.01. Likewise, exercise activity significantly decreased during the pandemic among men, *t*(158) = 2.70, *p* = 0.01.

### 3.2. Bivariate Associations between the Study Variables

Table 3 displays the bivariate correlations between all study variables for women and men, respectively.

In both genders, higher general and COVID-related stress levels were strongly associated with deteriorations in all three mood dimensions (all *p* < 0.001). However, the association between general stress and calmness was significantly stronger in men than in women, *z* = 1.85, *p* = 0.03. Exercise activity before the pandemic was unrelated to stress levels and mood among both women and men. Women with higher exercise activity during the pandemic reported lower COVID-related stress levels, more energy, and a more positive mood (all *p* < 0.01). Men with higher exercise activity during the pandemic reported more energy (*r* = 0.18, *p* < 0.05). Various gender differences were found in the associations of emotion regulation with the other variables. The positive association between COVID-related stress and emotion regulation was stronger in men than in women, *z* = 1.50, *p* = 0.001. Engaging in more emotion regulation was significantly associated with deteriorations in mood in men, but was unrelated in women (energy: *z* = 1.86, *p* = 0.03, valence: *z* = 2.44, *p* = 0.01, calmness: *z* = 1.69, *p* = 0.05). Furthermore, the association between exercise activity before the pandemic and emotion regulation was significantly different between women and men, *z* = −2.19, *p* = 0.01. All other comparisons were not significant.

### 3.3. Exercise Activity and Emotion Regulation as a Moderator of the Stress-Mood Relationship

Table 4 and Table 5 display the results of the hierarchical regression analyses for women and men, respectively.

For women, the hierarchical regression models explained between 14 and 36 percent of the variance in mood, with higher levels of explained variance in the general stress models compared to the COVID-related stress models. The analyses revealed no significant interaction effects between stress, exercise activity during the pandemic, change in exercise activity and emotion regulation. General and COVID-related stress were significantly associated with mood deteriorations in all models. While exercise activity during the pandemic was positively associated with energy (in both stress models) and valence (in the general stress models), change in exercise activity was associated with energy only. Emotion regulation was positively associated with energy in the general stress models, and negatively associated with calmness in the COVID-related stress models.

For men, the regression models explained between 16 and 48 percent of variance in mood, with higher levels of explained variance in the general stress models compared to the COVID-related stress models. Counter to the results in women, the three-way interaction between general stress, exercise activity during the pandemic and emotion regulation explained two percent of the variance in energy (ß = −0.17, *p* = 0.04). The plotted interaction in Figure 2 shows that if the level of general stress is high, participants with low exercise activity and high use of emotion regulation reported significantly less energy than all other participants. Furthermore, the simple slope tests showed that two pairs of slopes significantly differed from each other (see Table 6). Specifically, the slope of participants with low exercise activity and high emotion regulation (i.e., slope 3) significantly differed from participants with low exercise activity and low emotion regulation (i.e., slope 4), in the sense that the former group experienced a greater decrease in energy if stress levels were high. In contrast, participants with high exercise activity and high emotion regulation (i.e., slope 1) significantly differed from their counterparts with low exercise activity and high emotion regulation (i.e., slope 3), with the latter group experiencing a greater decrease in energy if stress levels were high. In the other models, general and COVID-related stress were negatively associated with all mood dimensions. Exercise activity during the pandemic and change in exercise activity were positively associated with energy only. Emotion regulation was negatively associated with valence and calmness in the COVID-related stress models and with men’s energy in the COVID-related/exercise during the pandemic model.

## 4. Discussion

The aim of the present study was to examine the three-way interaction between stress, exercise activity and emotion regulation on mood in physically active young adults during the early phase of the COVID-19-pandemic in spring 2020, while accounting for gender differences. The present study adds to the current literature, as we examined for the first time whether an adaptive change in exercise activity in response to the COVID-19 pandemic would buffer against stress-related mood deteriorations. Our results reveal no significant stress-buffering effects of physical activity and emotion regulation on mood in women. In men, exercise activity buffered the negative effects of general stress on energy in the case of high emotion regulation, but less so in those with lower emotion regulation. If the level of general stress was high, male participants with low exercise activity and high use of emotion regulation reported significantly less energy than all other male participants. All the other models revealed no significant stress-buffering effects of exercise activity or moderating effects of emotion regulation on mood. The corresponding hypotheses will now be discussed in turn.

In line with our first hypothesis, perceived stress was negatively associated with mood in both men and women. General stress levels in the present study were remarkably higher than norm values for gender- and age-matched control groups outside the COVID-19 pandemic (for comparison see [76]). In contrast, international comparisons to students during the COVID-19 pandemic revealed that the German sports science students in the present sample experienced less stress than French [82], Polish [83], and American students [84]. In addition, the negative associations between stress and mood were stronger for general stress than for COVID-19 related stress. Given previous reports on high stress levels and mental health challenges faced by students in general [85,86,87,88], these findings suggest that the student population might be susceptible to experiencing poor well-being and mental health consequences, regardless of the COVID-19 pandemic. In line with previous findings [61,62], this might be particularly true for women. In the present sample, women experienced more general and COVID-related stress, felt less energetic and calm, and engaged in more emotion regulation than men.

Supporting our second hypothesis, exercise activity significantly decreased by on average nearly an hour per week during the pandemic in both men and women, which was accompanied by deteriorations in energy. In line with the literature on the beneficial effects of exercise activity on well-being [23], our results show that individuals who maintained or initiated exercise activity during the pandemic experienced more energy, suggesting exercise activity as an effective coping resource [7]. However, we acknowledge that our cross-sectional data cannot be interpreted causally. In other words, it cannot be ruled out that participants with more energy were more motivated to engage in exercise activity.

One possible mechanism through which exercise activity is assumed to increase well-being is by enhancing emotion regulation [44,45,46,47]. Contradicting our hypothesis, only men’s exercise activity before the pandemic, but not during the pandemic, was positively associated with the use of emotion regulation. We considered emotion regulation as a coping resource which would result in better well-being [79,89,90]. Specifically, four of the six strategies assessed (i.e., reappraisal, reflection, distraction, and social sharing) have been shown to increase positive affect [79,90,91,92], while the remaining two (i.e., suppression, rumination) have yielded mixed results, albeit more evidence for negative effects on well-being [57]. In our study, the use of emotion regulation was associated with higher stress levels and mood deteriorations. In the literature, reappraisal shows the most robust positive associations with well-being [57,90], but was used least often in the present sample (women: *M* = 3.36, *SD* = 1.70; men: *M* = 2.92, *SD* = 1.60). This might (partly) explain why emotion regulation as a resource to cope with stress appeared not to be successful. Rather, our results indicate that higher use of emotion regulation is an indicator of poor mental health. Following this argumentation, it is not surprising that emotion regulation did not buffer against stress-related mood deteriorations either, although it contradicts our third hypothesis. Importantly, most studies of emotion regulation (e.g., [79,89]) examined the broad impact of emotion regulation on well-being without considering how stressful life events might exacerbate the impact of emotion regulation on outcomes. For instance, distraction—as the most used emotion regulation strategy used in the present sample (women: *M* = 4.84, *SD* = 1.55; men: *M* = 4.49, *SD* = 1.66)—and rumination have been shown to be consistently associated with negative affect in the face of daily stressors [92,93] (for an overview, see [57]).

Additionally, contradicting our third hypothesis, neither exercise activity nor an (adaptive) change in exercise activity moderated the relationship between perceived stress and mood. This finding contradicts a body of literature on the stress-buffering of exercise activity [4,5,6]. Researchers have argued that the stress-moderating potential of exercise might be akin to the experiences of social integration (e.g., positive perception of social relationships, group affiliation and group cohesion) and social support (e.g., encouragement, comfort, impulses to participate [94]). Young people in particular, such as the sports science students in the present sample, who are used to engage in exercise in classes, might benefit more from enjoyable activities that involve interpersonal contact. However, the social distancing regulations during the COVID-19 pandemic meant that the participants were no longer allowed to engage in types of exercise activities that are presumably enjoyable or have social contact with peers. Rather, exercise activities such as jogging, hiking and workouts were conducted individually. Supporting this argumentation, Gerber et al. [95] found no stress-buffering effect of aerobic exercise in Swiss university students in comparison to the significant effects of ball sports and dancing. Therefore, they argue that highly stressed individuals should engage in activities that leave little time for rumination and involve social interaction. As such, the missing stress-buffering effect of exercise in the present study might be explained by the type of exercise and the peculiarities of the student sample.

Our fourth hypothesis on the three-way interaction of stress, exercise activity, and emotion regulation was not supported. The significant three-way interaction effect on men’s energy revealed a different pattern than the expected greatest mood deteriorations under high stress, low exercise activity and low emotion regulation. On the contrary, when participants with low exercise activity were faced with high stress, they experienced the greatest mood deteriorations in cases of high emotion regulation. Nevertheless, in cases of high stress and high emotion regulation, exercise activity did buffer against stress-related deteriorations in men’s energy, suggesting that exercise activity might facilitate the success of emotion regulation (cf. [44,45,46,47]).

Although this finding seems encouraging, the following limitations should be considered in the interpretation of the results in order to avoid overgeneralization of our findings. First, the moderation effect was small (approximately 2% of additionally explained variance), which is in line with most studies, with levels of explained variance ranging from <1% to 7% (for two-way interactions [95,96,97,98,99]; for a three-way interaction [100,101]). In field studies, the detection of interaction effects is difficult since main effects are accounted for before the interaction effects are taken into consideration, leaving limited amount of variance to be explained through the interaction term (e.g., [102,103]). This issue with the detection of small effect sizes might be particularly true in the present study for several reasons: to the best of our knowledge, this is the first study that has investigated the stress-buffering effects of exercise on positive mental health, embracing various dimensions of well-being. Previous research has focused on negative health outcomes, such as burnout [100,101], depression [95,96], fatigue [35], and physical health complaints [97,98,104]; but see [36]. In addition, our sample was highly active and stress levels were increased compared to the general population [76], but were still relatively moderate compared to international students during the COVID-19 pandemic [82,83,84]. As such, it is likely that more healthy students were more willing to participate in the study, resulting in a selection bias. Simultaneously, findings cannot be generalized to the general population. While we were specifically interested in a highly active population due to their pronounced negative responses to exercise deprivation [12,15,16], in fact, the positive effects of (an increase in) exercise might be higher in the general population. Although higher exercise activity levels are considered to be even more health-enhancing, the greatest health benefits are thought to occur in the shift from inactivity to recommended levels of the WHO [105]. Second, our analyses focused exclusively on leisure time exercise (namely sports activities), as leisure time physical activity has been shown to have the greatest effects on mental health [23]. However, we did not differentiate type and intensity of exercise activities which might have confounded our results: ball sports and dancing (in contrast to aerobic exercise), as well as moderate intensity (in contrast to vigorous or light intensity), were shown to have greater stress-buffering potential (cf. [95,97,106]). Additionally, non-exercise activities (e.g., taking a walk, household) were not considered, although they contribute to an individual’s overall physical activity level [22]. Given the peculiarities of the COVID-19 pandemic, these activities might have gained greater importance in the contribution to mental health. Third, all data were derived from subjective—but validated—self-report measures and might have resulted in report biases, especially for the retrospective assessment of exercise activity before the pandemic. Although there is a disagreement as to whether or not self-report measures allow a valid assessment of exercise engagement (e.g., [107,108]), prior studies provided acceptable estimates of exercise in self-report measures [78,109].

Recent studies suggest that emotion regulation strategies should not be differentiated into adaptive or maladaptive per se. Rather, variably choosing between different strategies within a situation may be adaptive in daily life [56]. Exercise activity might be linked to a specific emotion regulation strategy, potentially the disruption of rumination through a cognitive “time-out” [110,111]. In this regard, the role of social interaction and social support in emotion regulation through exercise should be investigated. Future studies could make use of ecological momentary assessment (potentially including accelerometers) to longitudinally capture the idiosyncratic dynamics of emotion regulation, exercise activity and well-being in face of daily stressors.

## 5. Conclusions

In light of the negative psychological effects of the COVID-19 pandemic [1,2,3,112,113], maintaining or initiating exercise activity has been discussed as an important coping resource to counteract stress-related deteriorations in well-being [7]. Our results provide evidence that exercise activity was directly associated with more energy in both men and women, regardless of life stress or emotion regulation, although the effects were rather small-sized (cf. [11,72,114]). Nevertheless, exercise is more than being active. Through indirect effects, regular exercise activity provides coping resources via social support, self-efficacy, better sleep quality [115], and protects resilience during the COVID-19 pandemic [116]. Our findings partially confirm such indirect mechanisms for the stress-buffering capabilities of exercise activity with a particular focus on emotion regulation. We found support for the potential of exercise activity to counteract stress-related mood deteriorations by facilitating emotion regulation in men. In the case of high stress, low levels of exercise activity and high use of emotion regulation were associated with the greatest deteriorations in energy in men, while high levels of exercise activity reduced energy deteriorations under these circumstances. Accordingly, we cautiously conclude that the provision of (regular) training opportunities and promotion of exercise programs might be an effective strategy in overall health promotion during the COVID-19 pandemic, especially because we observed an overall decrease in exercise activity. Nevertheless, future research is warranted to elucidate the role of specific emotion regulation strategies (i.e., rumination, social support) in the direct and stress-buffering effects of exercise activity on well-being.

## Figures and Tables

**Figure 1 ijerph-18-07117-f001:**
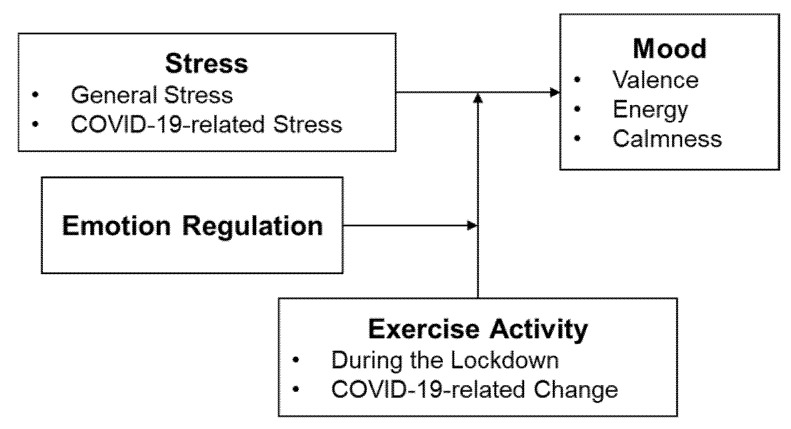
Three-way interaction model between stress, exercise activity, and emotion regulation on mood.

**Figure 2 ijerph-18-07117-f002:**
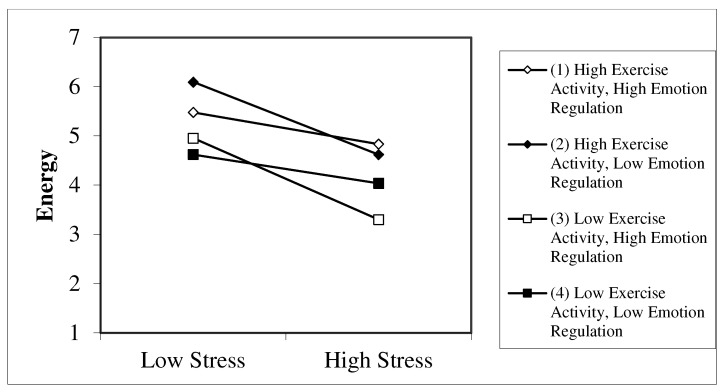
Three-way interaction between general stress, exercise activity during the pandemic, and emotion regulation on energy in men.

**Table 1 ijerph-18-07117-t001:** Frequencies (%) of type of sports mentioned before and during the pandemic.

Type of Sports	Before the Pandemic (*n* = 720)	During the Pandemic (*n* = 701)
Aerobic sports	33.6	54.5
Ball sports	21.0	28.3
Combat sports	26.6	2.5
Weight training	0.6	0.1
Gymnastics	8.6	13.8
Rehabilitation sports	5.7	0.3
Climbing/Bouldering	0.6	0.1

Note: Multiple responses possible.

**Table 2 ijerph-18-07117-t002:** Descriptive statistics and gender differences in study variables.

Measure	Total Sample		Women ^a^		Men ^b^		
	*M*	*SD*	*M*	*SD*	*M*	*SD*	*t*(362)
Energy (1–7)	4.53	1.41	4.39	1.46	4.71	1.33	2.15 *
Valence (1–7)	4.70	1.38	4.61	1.41	4.80	1.34	1.30
Calmness (1–7)	4.55	1.41	4.32	1.41	4.77	1.38	2.70 **
General Stress (0–40)	15.95	5.31	17.08	4.71	14.50	5.68	−4.64 ***
COVID-19 Stress (1–7)	3.81	1.17	4.04	1.05	3.51	1.25	−4.28 ***
Exercise Activity before the Pandemic (min/week)	255.66	280.63	243.71	297.47	271.07	282.26	0.92
Exercise Activity during the Pandemic (min/week)	201.78	231.70	192.02	210.27	214.37	256.84	0.91
Emotion Regulation (7–49)	23.48	5.04	24.28	4.68	22.44	5.30	−3.51 **

Notes: ^a^
*n* = 205 for women. ^b^
*n* = 159 for men; * *p* < 0.05, ** *p* < 0.01, *** *p* < 0.001.

**Table 3 ijerph-18-07117-t003:** Intercorrelations for study variables disaggregated by gender.

Variable	1	2	3	4	5	6	7	8
1. Energy	-	0.55 ***	0.55 ***	−0.45 ***	−0.37 ***	−0.11	0.18 *	−0.20 **
2. Valence	0.65 ***	-	0.71 ***	−0.63 ***	−0.37 ***	0.02	0.03	−0.32 ***
3. Calmness	0.50	0.70 ***	-	−0.70 ***	−0.39 ***	−0.06	−0.04	−0.36 ***
4. General Stress	−0.49 ***	−0.59 ***	−0.58 ***	-	0.48 ***	0.06	0.01	0.38 ***
5. COVID-19 Stress	−0.31 ***	−0.43 ***	−0.40 ***	0.41 ***	-	0.12	−0.06	0.30 ***
6. Exercise Activity Before the Pandemic	0.02	0.04	−0.04	0.03	−0.09	-	0.52 ***	0.17 *
7. Exercise Activity During the Pandemic	0.30 ***	0.20 **	0.12	−0.13	−0.23 ***	0.46 ***	-	0.13
8. Emotion Regulation	−0.01	−0.08	−0.19 **	0.30	0.15 *	−0.06	−0.03	-

Notes. The results for the female sample (*n* = 205) are shown below the diagonal. The results for the male sample *(n* = 159) are shown above the diagonal; * *p* < 0.05, ** *p* < 0.01, *** *p* < 0.001.

**Table 4 ijerph-18-07117-t004:** Hierarchical regression results for mood among women.

	General Stress × Exercise Activity During		COVID-19 Stress × Exercise Activity During		General Stress × Exercise Activity Change		COVID-19 Stress × Exercise Activity Change	
Energy	ß Final	ΔR^2^	ß Final	ΔR^2^	ß Final	ΔR^2^	ß Final	ΔR^2^
Step 1:		0.32 ***		0.15 ***		0.32 ***		0.17 ***
Stress	−0.52 ***		−0.26 ***		−0.51 ***		−0.27 ***	
Exercise	0.25 ***		0.27 **		0.25 ***		0.29 ***	
ER	0.15 *		0.04		0.14 ***		0.01	
Step 2:		0.01		0.01		0.01		0.01
Stress × Exercise	0.05		0.02		0.03		−0.04	
Stress × ER	−0.06		0.09		−0.08		0.06	
ER × Exercise	−0.01		0.05		−0.04		0.02	
Step 3:		0.00		0.00		0.00		0.01
Stress × Exercise × ER	0.06		−0.04		0.02		−0.12	
Total R^2^		0.30 ***		0.14 ***		0.30 ***		0.16 ***
**Valence**								
Step 1:		0.37 ***		0.19 ***		0.37 ***		0.20 ***
Stress	−0.61 ***		−0.40 ***		−0.61 ***		−0.40 ***	
Exercise	0.14 *		0.12		0.11		0.13	
ER	0.11		−0.03		0.10		−0.04	
Step 2:		0.01		0.01		0.01		0.01
Stress × Exercise	0.08		−0.01		0.06		−0.06	
Stress × ER	−0.09		−0.03		−0.10		−0.04	
ER × Exercise	−0.01		0.05		−0.04		0.04	
Step 3:		0.00		0.00		0.00		0.00
Stress × Exercise × ER	0.02		−0.05		−0.01		−0.08	
Total R^2^		0.36 ***		0.17 ***		0.36 ***		0.18 ***
**Calmness**								
Step 1:		0.34 ***		0.18 ***		0.34 ***		0.18 ***
Stress	−0.57 ***		−0.37 ***		−0.55 ***		−0.36 ***	
Exercise	0.06		0.05		0.08		0.12	
ER	−0.01		−0.17 *		−0.02		−0.18 **	
Step 2:		0.00		0.01		0.01		0.01
Stress × Exercise	0.04		−0.02		0.08		0.02	
Stress × ER	0.02		−0.04		−0.01		−0.05	
ER × Exercise	0.01		0.04		−0.03		0.00	
Step 3:		0.00		0.01		0.00		0.01
Stress × Exercise × ER	0.02		−0.10		−0.02		−0.13	
Total R^2^		0.32 ***		0.16 ***		0.32 ***		0.17 ***

Note. * *p* < 0.05, ** *p* < 0.01, *** *p* < 0.001.

**Table 5 ijerph-18-07117-t005:** Hierarchical Regression Results for Mood Among Men.

	General Stress × Exercise Activity During		COVID-19 Stress × Exercise Activity During		General Stress × Exercise Activity Change		COVID-19 Stress × Exercise Activity Change	
Energy	ß Final	ΔR^2^	ß Final	ΔR^2^	ß Final	ΔR^2^	ß Final	ΔR^2^
Step 1:		0.24 ***		0.17 ***		0.28 ***		0.20 ***
Stress	−0.41 ***		−0.28 **		−0.43 ***		−0.26 **	
Exercise	0.32 **		0.32 **		0.39 ***		0.36 ***	
ER	−0.08		−0.16 *		−0.04		−0.15	
Step 2:		0.01		0.00		0.01		0.01
Stress × Exercise	0.01		0.14		0.07		0.16	
Stress × ER	−0.04		0.02		−0.01		0.01	
ER × Exercise	−0.14		−0.14		−0.16		−0.12	
Step 3:		0.02 *		0.02		0.02		0.01
Stress × Exercise × ER	−0.17 *		−0.20		−0.14		−0.13	
Total R^2^		0.24 ***		0.16 ***		0.27 ***		0.19 ***
**Valence**								
Step 1:		0.41 ***		0.19 ***		0.40 ***		0.19 ***
Stress	−0.59 ***		−0.30 ***		−0.59 ***		−0.30 ***	
Exercise	0.05		0.04		0.04		−0.02	
ER	−0.11		−0.25 **		−0.10		−0.23 **	
Step 2:		0.01		0.01		0.01		0.00
Stress × Exercise	−0.09		0.06		−0.07		0.02	
Stress × ER	−0.03		−0.05		−0.04		−0.04	
ER × Exercise	−0.01		−0.03		−0.05		−0.02	
Step 3:		0.00		0.00		0.00		0.00
Stress × Exercise × ER	0.04		0.07		−0.01		0.07	
Total R^2^		0.39 ***		0.16 ***		0.39 ***		0.16 ***
**Calmness**								
Step 1:		0.49 ***		0.22 ***		0.49 ***		0.22 ***
Stress	−0.65 ***		−0.32 ***		−0.65 ***		−0.33 ***	
Exercise	0.00		−0.01		0.01		−0.04	
ER	−0.12		−0.26 **		−0.11		−0.27 **	
Step 2:		0.01		0.01		0.01		0.01
Stress × Exercise	0.01		0.08		0.07		0.10	
Stress × ER	−0.08		0.03		−0.08		0.04	
ER × Exercise	−0.01		0.02		−0.04		0.04	
Step 3:		0.00		0.00		0.00		0.00
Stress × Exercise × ER	−0.04		−0.05		−0.05		−0.04	
Total R^2^		0.48 ***		0.19 ***		0.48 ***		0.19 ***

Note. * *p* < 0.05, ** *p* < 0.01, *** *p* < 0.001.

**Table 6 ijerph-18-07117-t006:** Differences between slopes depicted in Figure 2.

Pair of Slopes	Slope Difference	*t*	95% Confidence Interval
(1) and (2)	0.07	1.38	(−0.03, 0.18)
(1) and (3)	0.09	2.18 *	(0.01, 0.17)
(1) and (4)	−0.01	−0.12	(−0.09, 0.08)
(2) and (3)	0.02	0.33	(−0.08, 0.11)
(2) and (4)	−0.08	−1.29	(−0.20, 0.04)
(3) and (4)	−0.09	−1.99 *	(−0.19, −0.00)

Notes. (1) = high exercise activity, high emotion regulation; (2) = high exercise activity, low emotion regulation; (3) = low exercise activity, high emotion regulation; (4) = low exercise activity, low emotion regulation; * *p* < 0.05.

## Data Availability

The data presented in this study are available on request from the corresponding author. The data are not publicly available due to the lack of participants’ consent to publish data.

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
