# Peer review of "Direct and Stress-Buffering Effects of COVID-19-Related Changes in Exercise Activity on the Well-Being of German Sport Students"

_ijerph, 2021, doi:10.3390/ijerph18137117_

Round 1
Reviewer 1 Report
It is my pleasure to give my insights and highlight some recommendations and commentaries to the manuscript, considering my expertise in the field. I really appreciate this opportunity to discuss the evidence and I hope my contributions could serve to improve the final version of the study.
Manuscript titled “Direct and stress-buffering effects of COVID-19-related changes in exercise activity on the well-being of German sport students”, deal an important issue of COVID-19 pandemic. The present study investigated the interaction between stress, exercise activity (EA) and ER on mood.
I believe that although the authors have carried out an interesting and amazing study and the content of the document has high potential and may be of interest to the field of study, there are some details that could be improved in order to increase its relevance, clarity, application practical and overall quality.
This article is interesting and suitable with the remit and purpose of the journal even if some revisions needed to be solved before advice publication.
In the abstract please add the scientific/ clinical relevance of this study.
Please improve the introduction section to help better readers understanding and to strengthen your work. Also, please update your work based on the current literature.
Please add a sentence regarding the physiological benefits of home-based exercise during pandemic.
Please add a sentence regarding the athletes’ negative effects during the COVID-19 forced stop period (decline in maximal oxygen consumption (VO2max), loss of endurance capacity, loss of muscle strength and mass, decrease joint lubrification, aerobic capacity impairs general performance.
Please add a sentence regarding the adapted physical activity to ensure the physical and psychological well-being of covid-19 patients and the importance of the movement during the forced rest period to counteract the psychological disorders.
Please provide the kind of sport the participants made in the section materials and methods.
Please provide more details regarding the ethical commitments that approved the study.
In the conclusion section please highlight better the scientific/clinical relevance of your work. Please provide a clear “take-home message” of the importance of this paper in the scientific community.
Reviewer 2 Report
Your paper has some potentially interesting points. However, it is filled with speculations which aren’t supported by your results. I feel as though you are reaching quite a bit when making claims throughout your discussion. Further, you found no novel data. Your results which go against your hypothesis could be of interest, however there are too many confounding variables to draw adequate conclusions.
Abstract
Assuming Mage means mean age but not clear you should specify this
Introduction
You start talking about stress and shift your focus to COVID-19 perhaps move that first line down the paragraph
You flip between present and past tense
Line 40: I believe you mean public not publish
Line 48: Explain why it is imagined…not sure why that is necessary or added.
Line 59: Define what you mean by affect and activation
You need to better define buffer if you repeated use it.
The first aim of the study was to investigate exercise as a protective coping resource in the current COVID-19 pandemic through its direct and stress-buffering effects on well-being. Your aims as listed in your intro are not very specific to what you are actual doing
Line 77 this sentence seems really out of place
The last paragraph of your introduction reads more like a discussion
Your flow chart does not help the reader
You never mention how you define “mood” until the flowchart.
Methods:
If you were going to compare gender differences at two time points before and during COVId-19 a 2 x 2 ANOVA would be more appropriate.
Results:
Any psychological benefit being moderately active vs. low or high?
Did mode of exercise make a difference?
Did the intensity of exercise make a difference?
Were any of these students, student athletes?
Could substance abuse be a confounding factor?
Discussion:
Line 326 I it’s not surprising subjects were more stressed during a pandemic but you are using a different study that occurred 5 years earlier. There a lot of other scenarios outside of COVID-19 that could lead to differences between those groups of subjects.
Line 335: That is not the focus of your particular study. You have no non-student control to make this claim.
Line 344: According to your study you didn’t find that. Energy levels alone do not mean poor mental health. There are a lot of other confounding variables during a lockdown that can lead to this state.
How well do you believe your subjects could recall their exercise habits before COVID-19?
Conclusion:
According to your data exercise did not have much of an effect and therefore how can your conclusion be that implementing an exercise protocol may be useful?
Reviewer 3 Report
The authors of the manuscript ‘Direct and Stress-Buffering Effects of COVID-19-Related Changes in Exercise Activity on the Well-Being of German Sport Students’ focused on understanding the role of exercise on stress and the overall wellbeing of sports students.
The authors recruited 366 participants between 18-39 years to address their aim, including males and females. The study design is rigorous with well-established Questionnaire tools. The authors have done an excellent job in conducting this study as well as writing components of manuscripts. The tables and figures are self-explanatory, and the conclusion reflects the hypothesis and results.
The only minor change could be,
Table 1- please add “combined (Women and Men)” at the top of the first 2 columns i.e. M and SD.
Author Response
Thank you very much for taking the time to review our manuscript.
We have added “Total sample” as a heading of the first two columns in the former Table 1, now Table 2 (see page 6).
Round 2
Reviewer 2 Report
Congratulations on improving your manuscript.
You have significantly improved the clarity of your writing and have addressed most of my concerns.